# Improving the Economic Sustainability of the Fashion Industry: A Conceptual Model Proposal

**Dejana Nikolic \* and Milica Kostic-Stankovic**

Faculty of Organisational Sciences, University of Belgrade, 11000 Belgrade, Serbia;
milica.kostic-stankovic@fon.bg.ac.rs
\* Correspondence: dejana.nikolic@fon.bg.ac.rs

**Abstract:** This study aimed to define the specific relationships between fashion style preference on the one side and product, promotion, word of mouth (WOM) and fashion lovers' behavior on the other side during the COVID-19 crisis. Structural equation modelling (SEM) analysis was employed to unveil the mutual relations of the two and to verify the proposed conceptual model. The conceptual model was tested based on the answers of 642 respondents. The preference for a specific fashion style proved to have an impact on the product, promotion, WOM and fashion lovers' behavior during COVID-19. Moreover, those elements have an impact on the frequency of fashion apparel consumption. Based on these results, fashion companies can tailor their activities in line with the predominant style of their fashion apparel to improve their economic sustainability during the post-COVID-19 era.

**Keywords:** fashion industry; economic sustainability; structural equation modelling; conceptual model; COVID-19

## 1. Introduction

Creative industries—which encompass industries such as film, music and fashion—have been hit extremely hard by the COVID-19 pandemic [1]. The creative industry that we focus on in this paper is the fashion industry. As COVID-19 spread, restrictions were imposed, leading to brick-and-mortar stores being closed and people staying at home [2]. This led to a momentary decline in value or even the complete ceasing of people shopping for clothes. Accordingly, problems arose for both companies engaged in selling fashion apparel as well as companies engaged in its production [3]. There was and is an urgent need for fashion businesses to adapt and try to overcome this sudden crisis [4]. As with many other creative industries, the fashion industry faced tremendous changes, difficulties, and challenges during the COVID-19 pandemic. Firstly, it is important to highlight the fact that at the start of the crisis, fashion was not a top priority for consumers [5]. People were occupied with preserving their health and the health of their families, buying food and other essentials, preserving their jobs, and getting used to working from home. All of this forced consumers to rethink their consumer needs and habits. These drastic, turbulent, immediate changes made even large retail companies such as Neiman Marcus and JC Penney file for bankruptcy, while the stock prices of giants such as Adidas and Nike dropped by 25% [6]. The Business of Fashion and McKinsey released a report examining the future of the USD 2.5 trillion global fashion industry. The report documented a staggering 90 percent decline in profit across the industry in 2020 during the COVID-19 pandemic. This marked the worst year on record; global fashion sales declined 15 to 30 percent in 2020 compared to 2019. Looking at specific regions, Europe saw a 22 to 35 percent decline in sales during 2020, the US saw a 17 to 32 percent decline, and China saw a 7 to 20 percent decline [7].

It is clear the fashion industry had to adapt to different consumer behaviors during the pandemic [8], especially as after only the first few months, the need and desire for the consumption of fashion items then increased [9].

Due to lockdown measures, consumers were not able to purchase fashion apparel in stores, which was, prior to the pandemic, the most common means of apparel shopping. Both consumers and retailers had to adapt: retailers had to offer online shopping options for fashion apparel and consumers had to accept the alternative means of shopping [10].

For some consumers, this transition in the way they shopped for clothes was nothing new, because they had already engaged in online shopping prior to the pandemic, but for the majority, this switch involved a substantial change in their shopping and consumer behavior [11]. A report by McKinsey [5] indicates that although there was a visible channel shift within the fashion industry, the online sales levels could not make up for the loss of in-store sales. The other change that hit consumers was in terms of fashion style. During the lockdowns, people had to stay home and work from home, meaning they had no need to dress in business wear or formally; instead, casual dress became the norm. Market research indicates that people also became more conscious regarding their fashion apparel consumption. During the lockdowns, consumers had time to think about what they love to wear and reconsider their fashion style. The desire to express themselves via their favorite fashion style, after a certain time under the pandemic, became highlighted in fashion lovers' mindsets. The reason for this can be assumed to be a lack of entertainment and extreme impatience towards the end of the pandemic, along with the restarting of special occasions [11]. The pandemic pushed the fashion industry to develop different management solutions as well as to strategically plan marketing activities that would enable retailers to understand consumers and react in the right way at the right moment. All of this caused a real demand shock in the overall fashion system [12]. However, leading fashion experts believe a fashion resurgence may be just around the corner, and it is believed that the scenario will probably be similar to that seen in the roaring twenties, with people soon going wild for fashion [13]. This is typical for luxury fashion brands, and a sudden rise in sales was noted in the luxury fashion market during the pandemic, compared to 2019 [14].Taking all the above into account, it can be concluded that the COVID-19 pandemic led to substantial changes for fashion companies, meaning they had to adapt to the changed environment and consumers [15,16]. Accordingly, it was of high importance for managers of companies operating in the fashion industry to understand how consumers' behavior had changed and what consumer preferences were making an impact, to anticipate consumer behavior and to improve their working process [15].

Taking into account previous research, it is important to highlight Badaoui, Lebrun, and Bouchet's [17] study that focused on the influence of clothing style identification on adolescents' brand consumption behavior. The sample consisted of 1063 adolescents. The researchers created a structural equation model, and the results indicated that the clothing brands adolescents preferred depended on the clothing style with which they identified. Meanwhile, Newman, and Patel [18] investigated variations in performance from a strategic and holistic view of fashion retailers' brand images, and assessed the adoption and orientation of quality marketing. Their survey was based on using key image attributes as a vehicle to understand customers' perceptions of the retailers under study. Survey data were collected from a sample of 300 typical customers who were set the task of ranking image attributes. A factor analysis of the customer responses indicated that Gap customers were offered lifestyles in advertising that were out-of-step with the merchandise on the shelves. This reflected a failure on the part of the retail management to satisfy the target market. Moreover, to investigate the optimal style launch strategy for a fashion firm, Zhou, Zhang, Gou, and Liang (2015) [19] proposed a two-period pricing model for a fashion product firm. Their main work was to illustrate which condition and which of their proposed strategies was the firm's best choice, as well as to determine the firm's optimal retail prices over the two periods under study. The results showed that a fashion firm's optimal launch strategy is mainly determined by the production cost and the consumers' mental book value [19]. It can be concluded that although many authors in their research have emphasized the importance of fashion style and certain elements of the marketing mix, to date no integrated research has been conducted that connects the influence of

fashion style and products and promotion. When it comes to the fashion industry and its specificities, it is of high importance to treat fashion style as a distinct term. In our study, we primarily focused on fashion style and how preferences for specific fashion styles shape fashion lovers' behavior. Fashion style can be described as the way that an individual expresses themselves through aesthetic choices, such as their clothing and the way they put an outfit together [20].

This exploratory study aims to provide a comprehensive conceptual model that sheds light on the importance of fashion style preference and its impact on products, promotion, WOM and fashion lovers' behavior during COVID-19, and their integrated impact on the frequency of consumption. The significance of this topic for the market lies in the fact that national and international interest in the Serbian fashion industry rose when experts recognized that this industry had exhibited a consistent recovery trajectory since 2007, after the country transitioned to a market economy [21]. Today, this industry is one of the most crucial creative industries in the Serbian economy.

The paper is organized as follows: the next section reviews the current literature on the impact of fashion style preference on consumer behavior and how consumer behavior related to fashion apparel consumption changed during the COVID-19 pandemic, while Section 3 focuses on the proposed conceptual model and the rationale for its construction. The next section covers the research methodology. Section 5 presents the obtained results and their managerial implications, while the paper finishes with a discussion and conclusion in Section 6.

## 2. Theoretical Background

### 2.1. Impact of Fashion Style Preference on Apparel Production and Consumer Behavior

Fashion items are unique consumer products characterized by short life cycles, high demand volatility, low sales predictability, and impulsive purchases [19]. Knowing that fashion companies have always faced the critical challenge of designing products that fit consumers' needs, the fashion industry lives on the cutting edge of design, continually reinventing itself while staying loyal to concrete fashion styles [20]. In the production of clothing, experience is often used to estimate the complete production process, which is subjective and lacks scientific basis [21]. To date, to our knowledge, no research has been conducted to link fashion style preference, apparel production and fashion lovers' behavior.

The importance of understanding the preferences of fashion lovers for the growth of the fashion industry may be fundamental. Shim and Kotsiopulos [22] suggested that retailers and manufacturers should collect information on the fashion style needs of specialty markets in order to produce apparel products that would yield greater satisfaction within the designated markets. Namely, there should be a congruence between fashion style and fit. Their study is important as it signals that fashion style alongside anthropometric differences should be taken into account in the production process [22].

Another good example of the important link between the fashion industry and fashion style preference can be found in the 1990s, when purely "street-born" fashion styles emerged in the young women's casual fashion market in Tokyo. This phenomenon revealed the limitations of the conventional framework of quick response (QR) approaches, which have long been implemented by Japanese fashion houses. These QR approaches were found to be incapable of responding to the need for fresh fashion designs for different fashion styles in the extremely volatile and fast-moving streets of Tokyo's casual fashion scene [23].

Fashion style has been shown to be one of the factors that impacts clothing purchasing decisions [24]. In the present study, we also analyzed some other elements, such as country of origin, positive image, fashion shows, promotion, WOM and COVID-19 behavior changes (change in shopping frequency and change in shopping place habits). When it comes to the countries of origin of the fashion brands, we aimed to investigate if there is any difference in behavior between those who prefer foreign/domestic fashion brands and those who do not. Further, positive image is interpreted as the willingness to wear branded clothing in order to contribute towards the creation of a positive image of oneself,

so this was measured too. Fashion shows are seen as a significant tool in marketing communication, so special attention is paid to the preferences of those who follow fashion shows and which kind of fashion shows they are attracted to. Additionally, promotion is one of the main tools in fashion marketing communication, so we paid special attention to the analysis of the perception of different channels of promotion. WOM in creative industries plays a significant role, so we included it in our analysis. When it comes to COVID-19, it is presumed that consumer behavior has changed; knowing this, certain aspects were included in the model: COVID-19 shopping frequency changes, as well as habit changes with regards to shopping locations during COVID-19.

Holmlund et al. [25] showed that for mature women, fashion style is one of the significant factors they consider when purchasing clothes. De Kervenoael et al. [26] provided evidence that besides functional apparel values (i.e., cost, quality, guarantee, warranty, etc.), intuitive factors such as fashion style are highly important when buying fashion apparel. Badaoui, Lebrun and Bouchet [17] analyzed the role of fashion styles as part of adolescent consumers' behavior. They created a structural equation model that explored how adolescents' sensitivity to media influence varies according to the fashion style and the mutual relationship between fashion styles and preferred brands. Their results indicated that both relationships exist.

Newman and Patel [18] highlighted that consumer behavior in the fashion marketplace is characterized by impulse purchasing and fickle customers. Fast fashion brands (such as Zara, H&M, Topshop and Mango) are aware of this, and adopt the strategy of constantly renewing their product ranges with fashion styles favored by their target groups. Namely, a fashion product firm/brand launches several fashion styles with various characteristics for each season [19].

An individual who is passionate about a fashion style and demonstrates passion-driven behaviors consequently invests time and money in that type of fashion. Content analysis carried out in the study of Louriero et al. [20] showed that self-identification with a fashion style is a highly important factor that motivates the consumer to make purchases. This study indicates the role of fashion style in shaping consumer behavior and reaches the conclusion that, in order to achieve and maintain competitive advantage, fashion companies should pursue a strategy that aligns closely with the customer fashion style preferences of their target groups.

*2.2. Consumer Behavior Change Amid COVID-19 Pandemic*

In the current highly competitive marketing environment in the fashion industry, to maximize long-term performance, the anticipation of consumers' future behavior and purchase intentions is a key strategic asset that must be observed and cherished [4]. Therefore, it is of utmost importance to observe and adapt to the changed consumers of fashion apparel. In the next paragraphs, we will review some noticed behavioral changes.

The first noted behavioral change was channel switching. Namely, as stores were closed, both consumers and retailers had to turn to alternative ways of shopping and selling, e.g., to e-commerce. According to the Coronavirus Response Survey conducted in the US, 73% of consumers said that the coronavirus experience will change the way they shop in the future, while 64% indicated that they will buy more clothing online in the future [27]. Both fast fashion and luxury fashion had to adapt quickly. It is believed that the losses in the fast fashion industry were lower than in luxury fashion, as along with closed stores, duty free shops were closed and international travel was ceased, which impacted sales of luxury goods [28].

Purchasing channels also changed. To strengthen the bond between customers and fashion brands, customer relationships are of prime importance [29]. Therefore, the use of social media for marketing communications seems to be the most apt medium, especially during crises such as the COVID-19 period. Many fashion brands, even fashion luxury brands, have created Twitter accounts, Facebook pages, Instagram pages, and even TikTok accounts in order to adapt to the new climate and to create the best connections with

their consumers [30]. Even Vogue has acknowledged the communication channel switch, especially to TikTok, noticing that due to staying at home, TikTok became the outlet for self-expression and creativity and a place for finding fashion ideas [31]. Another good example of a switch in communication channels is the fact that the Paris department store Galeries Lafayette launched live shopping sessions with leading luxury brands [32].

A report from Sales Intelligence [33] pointed out that there was a visible change in the clothing categories consumers were buying from during the COVID-19 pandemic. Namely, the popularity of sportswear and homewear increased, while interest in special occasion clothes such as dresses, skirts, and suits declined. Such a change could have been expected, as due to lockdowns, people were mostly at home and needed comfortable everyday clothes. Further, all special occasions and gatherings were cancelled as part of epidemiological measures, and there was thus no need for buying "fancy" clothes. As COVID-19 measures lessen and smaller gatherings are allowed, interest in special occasion clothes is expected to rise.

Another behavioral change that has been noticed is the switch to a circular economy and sustainability. Namely, during lockdowns, consumers had time to look into their wardrobes and decide whether they will continue wearing certain apparel or not [5]. Instead of throwing it away, consumers had three options: to rent it, to donate it, or to sell it. Fashion rental platforms recently emerged within the fashion industry, and for a short period of time received media attention, gained popularity among users, and attracted considerable financial investment [34]. These platforms allow users to rent clothes for a certain period at a certain price without owning them. On the other hand, a report by the Waste Resources and Action Program estimates that 67 million pieces of clothing and 22 million pairs of shoes will be disposed of by homes in the country post lockdown, with many of these going to charity shops [35]. Additionally, consumers turned to selling their apparel. Resale platforms such as Poshmark and Depop saw visible increases in activity in the COVID-19 period [36].

Bearing in mind all the noticed and documented changes in the fashion industry, marketing activities should adapt adequately to the changes in their consumers and environment.

## 3. Hypothesis and Model Development

Fashion companies are, especially now in the COVID-19 era, keen to identify factors that contribute to their consumers' interest in purchasing apparel products [37]. The model we herein propose aims at shedding light on how the preference of a fashion style contributes to the complex process of purchase decision making in the fashion industry. Fashion style preference was included in the model as the main moderator, as a previous study showed that fashion style preference has an important influence on consumers' perception of fashion product cues, fashion companies' marketing activities and brands [38].

The model itself consists of four blocks. The first encompasses preferences for particular fashion styles. Herein we observed individual preferences for six fashion styles: business, hip hop, retro, sports, casual, and rock and roll. Taking into account the complexity of the model itself, the second block covers products, promotion, and WOM. Marketing activities, particularly the concept of the 4Ps (price, product, promotion, placement), have long played important roles in marketing strategies in the fashion industry, and have been regarded as important variables that influence corporate equity value. Until now, most corporate activities have been conducted within the scope of 4Ps strategies, which help determine how to localize or standardize marketing mix factors if a global fashion company expands into a new market [39]. A fashion retail marketing strategy is closely related to a more general marketing strategy and works from a similar base; however, certain elements of the fashion industry dictate that the standard '4Ps' marketing framework is not sufficient as it concerns itself with products. Namely, the focus needs to be on market orientation, which requires a more detailed framework, and therefore an adaptation of the 4Ps model that goes beyond the product orientation of the 4Ps model is needed. There are a number of specific sub-elements to the fashion marketing mix that have been detailed by academics [40].

The fashion marketing mix refers to the 4Ps (product, price, placement and distribution and promotion) in the fashion industry. However, there are some typical sub-elements of the fashion marketing mix, such as country of origin (element-product), material quality (element-product), the fashion designer behind a product (element-product), offline shopping (element-placement), online shopping (element-placement), advertising through digital platforms (element-promotion), advertising through social media (element-promotion), advertising through TV (element-promotion), fashion shows (element-promotion), email marketing (element-promotion), etc. [41]. When it comes to certain elements of the fashion marketing mix, special attention is paid to some sub-elements of the 4Ps. Additionally, special attention is given to WOM, which has been found to be highly persuasive and thus extremely effective in influencing the consumption of products/services, especially in the fashion industry. This is due to WOM recommendations being typically spread by consumers who are perceived to have no personal interest in recommending a particular brand or product, which therefore renders such recommendations more credible in the fashion marketplace [42].

The third block aims to map consumer behavior changes due to COVID-19. Effects of COVID-19 on consumer behavior in general are obvious, so we dedicated the third block to discovering what these changes are in the fashion industry. Attention is paid to changes in the behavior of fashion lovers during the pandemic related to shopping habits and frequency change. The final block is the frequency of consumption. Based on the mutual relations between the blocks, we derived the following five hypothesis.

**Hypothesis 1 (H1).** *Preferences for fashion styles impacted the importance of product, promotion and WOM during the COVID-19 pandemic.*

A study by Kaiser [43] showed that individuals who exhibit a higher level of clothing interest usually spend a significant amount of time, money and energy on clothing selection. Wang [44] found that today's consumers are very well informed about fashion apparels, and if not, they can easily gain the necessary information when shopping. She also pointed out that those who are interested in particular fashion styles do not search for much additional information. Fornazarič and Toroš [45] showed that based on preferences for clothing color, consumer behavior differed. As consumers who are fans of a particular fashion style are meticulous, it is believed that such consumers will pay more attention to intrinsic and extrinsic aspects of the product, as well as to promotion and fashion shows.

**Hypothesis 2 (H2).** *Preferences for different fashion styles led to different apparel shopping habit changes during the COVID-19 pandemic.*

Fornazarič and Toroš [45] explored how threat appraisal, coping appraisal, and beliefs impact channel switching behavior, and found positive, statistically significant impacts. Tirtayasa et al. [41] found that shopping habits change and are differentiated based on lifestyle, fashion involvement, hedonic shopping motivation, impulse buying and other factors. Here, we hypothesized that consumers who prefer different fashion styles have different shopping behaviors.

**Hypothesis 3 (H3).** *Preferences for fashion styles impacted the frequency of fashion apparel consumption during the COVID-19 pandemic.*

Wang [44] found a relationship between work-time dress style and fast fashion shopping frequency. Namely, those who had to dress formally for work had a higher frequency of fashion apparel consumption. The same author found that those who were fans of sport and elegant style shopped less. Gupta et al. [46] found that, the more a consumer is a fan of a particular fashion style, the more their shopping frequency will decrease. Here we hypothesized that consumers who prefer different fashion styles have different apparel shopping frequencies.

**Hypothesis 4 (H4).** *The importance of product, promotion and WOM impacted the frequency of apparel shopping during the COVID-19 pandemic.*

Kovač, Palić and Tolić [47] showed that there is a positive correlation between the opinion that fashion apparel is an indication of status and shopping frequency. Mican and Sitar-Taut [48] showed that the most important factors which impacted apparel shopping during the COVID-19 pandemic were product characteristics, price, recommendations (online and offline), and product popularity. Accordingly, here, we hypothesized that consumers who pay more attention to product, promotion and WOM shop for fashion apparel more often.

**Hypothesis 5 (H5).** *Changes in apparel shopping habits impacted the frequency of apparel shopping during the COVID-19 pandemic.*

Research conducted in the UK showed that 46% of respondents indicated that their shopping habits changed due to COVID-19 [49]. The same study observed the frequency of apparel shopping in different channels. Namely, 55% of respondents stated that they had stopped shopping in store, while 50% stated that they had bought at least once a month online. McKinsey [50] predicted a decrease in the frequency of fashion apparel shopping due to COVID-19. Here we hypothesized that changes in apparel shopping habits due to COVID-19 had a statistically significant impact on the frequency of apparel shopping. The proposed conceptual model is graphically presented in Figure 1. The ovals present constructs in the model, while rectangles present individual variables in the model.

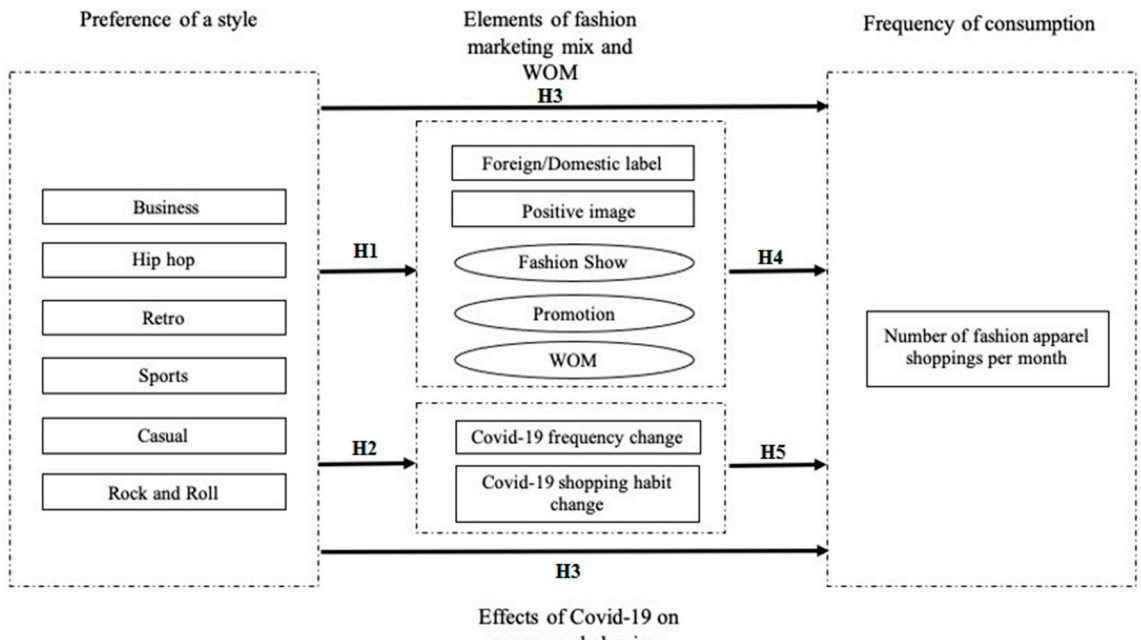

**Figure 1.** Proposed conceptual model.

## 4. Research Methodology

### 4.1. Procedure, Participants, and the Research Instrument

The survey was conducted online from 4th March until 10th May 2021 using the Google Forms service. At the time when the survey was open, as a part of measures to combat the COVID-19 pandemic, shopping malls and shopping galleries were closed in Serbia. The survey was distributed on Facebook groups and Instagram profiles related to fashion, students' groups, and on the personal profiles of the authors. The sampling method that we applied was convenience sampling, which is a nonprobability sampling

method. Participation was voluntary and anonymous, and respondents were informed that the data would be used for research purposes only. The COVID-19 pandemic has gone through different stages of development with different relative impacts, e.g., in relation to the geographical area. Serbia was one of the countries where quarantine lasted for most of 2021. Therefore, it is important to highlight the fact that it is believed that the period of administration of the survey, and the related restrictive measures may have influenced the responses obtained.

The survey consisted of five parts: demographic questions and sets of questions for each block. To measure fashion style preference, the respondent had to indicate his/her preference for a fashion style on a scale from 1 to 5. Based on conducted research by different Serbian media platforms, it was concluded that certain fashion styles are especially relevant for consumers in Serbia. According to Serbian popular media research, the six most represented fashion styles in Serbia are: business style, hip hop style, casual style, sport style, rock and roll style and retro style. The impact of preferences for these specific fashion styles was explored in the conducted research. To measure shopping frequency, the respondent was asked to enter the number of monthly purchases of fashion apparel they had made. To measure the remaining two blocks, we used similar questions to previous studies, such as Rahman and Kharb [51], Kovač, Palić and Tolić [47], and Wang [44]. Additionally, some questions were added by the authors. More details on the questions used in the survey are given in the sections below and in the Appendix A.

Bearing in mind that we used convenience sampling, we needed to check for the presence of common source bias (CSB). CSB indicates potential issues when a survey is used to measure both independent and dependent variables simultaneously, which might lead to inflated results [52]. To test for CSB in our data, we used Harman's single factor test. The test showed that one factor explained 23.004% of total variance, which is less than the threshold of 50%. Therefore, we can conclude that there was no common source bias in our data.

*4.2. Statistical Analysis*

To confirm or reject the devised hypotheses, we employed structural equation modelling (SEM) analysis. SEM is a statistical multivariate analysis which is based on the principles of factor analysis on one side, and multiple linear regression on the other [53]. Accordingly, the analysis can be used to reduce the dimensionality of the problem through the creation of latent variables or constructs and to later explore the relationship between the newly created variables. Bearing in mind the benefits of the analysis, its solid theoretical background, and the several software solutions available, its popularity has risen in the last 20 years.

To date, SEM analysis has been used in research related to fashion marketing and consumer behavior with regards to fashion apparel shopping. In an interesting study, Kim [54] tested a model of the factors which impact customer–brand resonance and fashion–brand experience using two rival brands. Giovannini et al. [55] aimed to model the factors that impact purchase intention and lead to brand loyalty in the sector of luxury fashion. In more recent studies, Kumar et al. [56] wanted to unveil the impacts of consumers' environmentally responsible fashion apparel purchase decisions, while Jain [57] explored the moderating role of subjective norms on the purchase intention of luxury fashion apparel among Generation Y consumers. Led by these examples of good practice, we were encouraged to use the SEM analysis.

## 5. Research Results

*5.1. Sample Characteristics*

In our sample, we had 2170 respondents. However, we wanted to be sure that we were studying those who are interested in fashion and take shopping for fashion apparel seriously. Therefore, in our study, we included only those who agreed or completely agreed with the statement "I observe myself as a fan of fashion". Out of the initial 2170 respondents,

only 642 expressed a form of agreement and were suitable for the study, as they declared themselves fashion lovers.

In the reduced sample, we had 118 (18.4%) males and 524 (81.6%) females. There was a visible gender disproportion in the sample. However, we did not treat the disproportion as biased, bearing in mind that females have been shown to be more prone to participating in online surveys [58] and that the topic of the survey is generally observed as a feminine one [59]. The respondents' average age was 30.99, with a standard deviation of 10.901 and a median of 29 obtained. When it comes to the highest completed level of educational attainment, most of the respondents had completed high school (39.4%), followed by those who had finished bachelor studies or their equivalent (37.5%), and those with a masters or PhD diploma (22.4%). Regarding their level of personal monthly income, most of the respondents indicated that they had no monthly income (27.1%); such a result was expected considering that the sample included a large percentage of students. Regarding those who had personal income, most of them indicated having above 700 EUR (22.6%), between 400 and 700 EUR (16.4%), and between 150 and 400 EUR (16.0%). Most of the respondents resided in Belgrade (52.6%), while the remaining were from other cities in Serbia, such as Novi Sad (8.6%) and others.

It can be concluded that our sample covered two segments of the millennial population: one consisting of highly educated individuals with an above-average income, and the other encompassing students in their final years of study.

### 5.2. Verification of Proposed Scales

Prior to conducting the SEM analysis, we tested the internal consistency of the proposed scales. To do so we used Cronbach's alpha, Average Variance Extracted (AVE), and Composite Reliability (CR). Cronbach's alpha measures the extent to which all items quantify the same concept [60]. It takes values between 0 and 1, where values above 0.7 are seen as acceptable [61]. The closer the AVE and CR are to 1, the better the internal consistency is, thus showing that the scale is more reliable. The acceptable level threshold for AVE is above 0.5, while for Composite Reliability it is above 0.7 [62].

The obtained metrics are provided in Table 1. As can be noted, for all three constructs in the model, the obtained Cronbach's alpha was above the 0.7 threshold. The same applied for the values of CR, which were also above 0.7. The value of AVE for the construct *Promotion* was slightly below the 0.5 threshold. Taking into account the same size and the number of items within the constructs, we can conclude that the proposed constructs were valid and suitable for SEM analysis.

**Table 1.** Cronbach's alpha, AVE, CR and the number of items per construct.

| Construct | WOM | Promotion | Fashion Shows |
|---|---|---|---|
| Number of questions | 4 | 8 | 3 |
| Cronbach's alpha | 0.740 | 0.832 | 0.837 |
| AVE | 0.568 | 0.465 | 0.755 |
| CR | 0.840 | 0.873 | 0.902 |

### 5.3. Validation of the Proposed Conceptual Model

The initial model had a solid fit to the data (Chi-square = 1182.620, df = 291, $p < 0.000$, RMSEA = 0.069, CFI = 0.867, TLI = 0.840, SRMR = 0.098). Therefore, to fine-tune and enhance our model, we used modification indices and removed all paths which were not statistically significant (whose critical ratio in absolute values was below 1.96).

The final model had a good fit to the data (Chi-square = 509.887, df = 268, $p < 0.000$, RMSEA = 0.038, CFI = 0.937, TLI = 0.923, SRMR = 0.056). The comparative indexes, Comparative Fit Index (CFI) and Tucker Lewis Index (TLI) were acceptable. The standardized root mean square residual (SRMR) was on the 0.05 threshold. Taking into account all the above presented results, it can be concluded that the final model can be used to draw conclusions [60].

The final model assessment is given in Table 2.

**Table 2.** Assessment of the final model: construct, predictors, obtained unstandardized and standardized coefficients, CR (* $p < 0.05$, ** $p < 0.01$), and the R square.

| Construct | Predictors | UnStd. Coeff | Std Coeff | CR | $R^2$ |
|---|---|---|---|---|---|
| WOM | Business style | 0.034 | 0.084 | 1.197 * | 0.016 |
| | Retro style | 0.037 | 0.095 | 2.271 * | |
| Promotion | Business style | 0.055 | 0.173 | 3.882 ** | 0.039 |
| | Sport style | 0.095 | 0.094 | 2.803 ** | |
| Fashion show | Business style | 0.037 | 0.112 | 2.675 ** | 0.025 |
| | Hip hop style | 0.048 | 0.110 | 2.709 ** | |
| Foreign/Domestic label | Casual style | 0.095 | 0.105 | 2.803 ** | 0.016 |
| | Business style | 0.041 | 0.072 | 1.969 * | |
| Positive image | Business style | 0.064 | 0.113 | 2.891 ** | 0.028 |
| | Retro style | −0.070 | −0.124 | −3.278 ** | |
| COVID shopping habit changes | Rock and roll style | 0.049 | 0.080 | 2.048 * | 0.012 |
| | Business style | 0.044 | 0.072 | 1.960 * | |
| COVID frequency changes | Hip hop style | 0.051 | 0.085 | 1.960 * | 0.007 |
| Frequency of shopping for fashion apparel | Hip hop style | −0.132 | −0.084 | −2.222 * | 0.103 |
| | Foreign/domestic label | 0.218 | 0.105 | 2.713 ** | |
| | Fashion show | 0.328 | 0.092 | 2.289 * | |
| | Positive image | 0.226 | 0.109 | 2.821 ** | |
| | COVID frequency | 0.515 | 0.200 | 5.314 ** | |
| | COVID means of shopping | 0.293 | 0.149 | 3.980 ** | |

Note: * $p < 0.05$, ** $p < 0.01$.

The construct *WOM* was measured using four factors regarding the importance of recommendations on purchase intention: material quality, fashion designer, fashion label, apparel itself. Two fashion styles proved to have a statistically significant impact on *WOM*: the business and retro styles. Both coefficients were weak, positive, and statistically significant. Therefore, the more the respondent was a fan of the business and retro styles, the more he or she was found to take into account fashion apparel recommendations.

The construct *Promotion* was quantified using eight factors regarding the importance of promotion on different channels on purchase intention: Internet portals, TV, print media, billboards, social network, organization of events, sponsorship of events, and direct marketing.

Two fashion styles proved to have a statistically significant impact on *Promotion*: the business and sport styles. As in the previous model, both coefficients were weak, positive, and statistically significant. The measured impact was such that the more the respondent was a fan of the business and sport styles, the more he or she was found to pay attention to promotion when making a purchase decision.

Bearing in mind that the research was dedicated primarily to fashion lovers, we paid special attention to the sub-element of promotion, *Fashion shows*, knowing it is assumed that fashion lovers have a strong interest in this sub-element. The construct *Fashion shows* was quantified using three factors regarding the importance of apparel appearing on fashion shows: on domestic fashion shows, on international fashion shows, and on fashion TV channels. Again, two fashion styles proved to have a statistically significant impact on *Fashion shows*: the business and hip hop styles. As in the previous models, both coefficients were weak, positive, and statistically significant. This result indicates that if a respondent

was a fan of the business and hip hop styles, he or she was found to take into account the trends shown on fashion shows when making a purchase decision.

Two styles proved to have an impact on the importance of the country of origin: the casual and business styles. The variable *Foreign/Domestic label* was assessed in the question "*I would rather buy a fashion apparel if the label was*" coded 1—Domestic, 2—It does not matter, and 3—Foreign. As in the previous models, both coefficients were weak, positive, and statistically significant. The fans of the business and casual styles tended to prefer the fashion apparel of foreign labels.

When it comes to the importance of buying the fashion apparel of famous brands to improve one's self-image, two styles proved to have a statistically significant impact: the business and retro styles. Interestingly, a preference for business style had a positive impact (0.037), while a preference for retro style had a negative impact (−0.070). Therefore, the fans of business style were found to believe that buying fashion apparel of famous brands leads to a positive self-image, while the fans of retro style did not share that opinion.

The results indicate that the fans of the rock and roll and business styles changed their shopping habits during the COVID-19 pandemic. One question aimed to assess how respondents' shopping behavior had changed, and this was coded the following: 1—I still buy fashion apparel only in stores, 2—I still buy fashion apparel both online and in store, 3—I started buying fashion apparel online, and 4—I started buying fashion apparel only online. As in the previous models, both coefficients were weak, positive, and statistically significant. The obtained coefficients indicate that the fans of these two styles changed their shopping habits and turned to online shopping during the pandemic.

Regarding the respondents' changes in the frequency of consumption during COVID-19, only a preference for hip hop style proved to have a statistically significant, positive, and weak impact. The fans of hip hop style tended to shop more often during the pandemic.

In the final model, the model of the frequency of the monthly consumption of fashion apparel, six predictors were identified: one fashion style (hip hop), certain sub-elements of the fashion marketing mix (*Foreign/Domestic label, Fashion shows* and *Positive image*), and both COVID-related statements. All predictors except hip hop had a positive, statistically significant impact. Those who turned to online shopping reported believing in enhancing their self-image through fashion apparel, following fashion shows, and preferring foreign labels, and these respondents had a higher fashion apparel consumption frequency than the other respondents.

What can also be concluded from Table 2 is that the observed $R^2$ values were quite low. In a way, such results are in line with previous research. For example, Cho and Workman [63] obtained low $R^2$ values when modelling consumer behavior while shopping for apparel.

Observing the results, we can see that only the preference for hip hop style had a direct impact on the frequency of consumption. What makes SEM analysis stand out is the possibility to observe the indirect effects. Therefore, in Table 3, we present the indirect effects of preferences for different fashion styles on the frequency of consumption. The only style preference which had no effect on the frequency of consumption was sports style. Being a fan of casual, business and rock and roll styles indirectly increased the frequency of consumption. On the other hand, being a fan of retro style decreased it. The observed coefficients were low and statistically significant on the level $p < 0.05$.

**Table 3.** Standardized and unstandardized effects of preferences for observed fashion styles on frequency of consumption.

|  | Casual Style | Retro Style | Business Style | Rock and Roll Style | Sport Style |
|---|---|---|---|---|---|
| Std | 0.011 * | −0.013 * | 0.041 * | 0.012 * | 0.000 * |
| UnStd | 0.021 * | −0.016 * | 0.049 * | 0.015 * | 0.000 * |

Note: * $p < 0.05$.

*5.4. Discussion and Managerial Implications*

Within the film industry, Desai and Basuroy [64] found that the higher the familiarity of and preference for a genre, the more reliance on other factors on the decision to watch a movie will be reduced. This can easily be translated to the fashion industry. The results of our study illustrate that preferences for different fashion styles have an impact on the importance of product, promotion and WOM. Namely, marketing activities and the optimal marketing mix can be determined in relation to a specific target market, segmented through the degree of preference for a particular fashion style. From our study, for example, we can see that fans of business style pay attention to WOM and five different sub-elements of the marketing mix. Therefore, companies who produce business style apparel should focus on active and detailed communication with their consumers.

An important managerial implication is also that the fashion style of an item of apparel is not enough to predict the importance of certain aspects of the fashion apparel to a potential consumer and the frequency of consumption. Looking at the parallel from the film industry, Gunter [63] states that the genre of the movie and genre preference can play an important part in the decision, but it does not operate on its own. The obtained results could be interpreted as suggesting that to a sole fan, a fashion style is not the only factor which determines the person's decision to buy certain clothes or to take into account information about it.

Considering the fact that during a pandemic, consumers are more emotional, sensitive and eager for direct communication, it can be assumed that a specialized communication strategy and the precise selection of marketing activities, in accordance with different affinities towards fashion styles, can lead to stronger connections between consumers and fashion companies, and this especially applies for fashion lovers. The presence of empathy and understanding the needs of special target groups of consumers during the COVID-19 pandemic, based on their preferences for specific fashion styles, can only lead to business improvements in the fashion industry, as well as to mutual satisfaction. Understanding the above and adapting marketing activities, and relying on preferences for certain fashion styles can be very important for fashion companies and retailers in general looking to improve their image, reputation, the loyalty of existing consumers and finally their sales during the COVID-19 and post-COVID-19 eras.

## 6. Concluding Remarks

Observing the pre- and post-COVID-19 periods, it is clear that fashion companies are eager to determine what creates interest among their consumers and what they pay attention to when buying certain types of clothing [43]. Recent research has shown that conventionally used segmentation criteria such as age, gender, and income are outdated, as they cannot portray the personalities and values shared by Millennials and Gen Z consumers [65]. Fast fashion companies are aware of this, and have increased the number of buying personas (ideal clients) from 2 to 3 to a staggering 10 to 15 [66]. Due to the lack of historical data, constantly changing fashion trends, and product demand uncertainty, demand forecasting is an important and challenging task in the fashion industry, which can lead to improvements in economic sustainability in the industry [67]. Fashion companies are trying their best to figure out which needs they should aim to solve for each persona and to elevate their customer experiences. One of the aspects that they are increasingly including in their strategies and segmentations is fashion style preferences. With access to seemingly endless information via the Internet, modern consumers are well-informed and interested in supporting fashion companies that reflect their fashion style and personality [68].

This study aimed to unravel the role of fashion style preference in fashion lovers' behavior during the COVID-19 period. Namely, on one side, we hypothesized that the importance of fashion style preference impacted the level of importance given to the product, promotion, WOM, and consumer behavior changes due to COVID-19, and the frequency of fashion apparel shopping. On the other side, we hypothesized that the importance given to product, promotion, WOM and consumer behavior changes due

to COVID-19 impacted the frequency of fashion apparel shopping. All of the paper's hypotheses (H1, H2, H3, H4 and H5) were confirmed, and all relationships, direct and indirect, were detected.

Now, more than ever, consumer behavior in the fashion industry is an agreement, something that looks like a type of friendship. Fashion companies need to listen to and understand the needs of their consumers in order to improve their economic sustainability, and especially the needs of consumers who are dedicated fans of the industry, in order to be able to meet and predict their demands, which can be interpreted as a core quality of great communication. The fashion business needs to understand changes in the mass market, on one hand, and to understand the typical behavior of their target audiences connected to their preferences for fashion styles on the other hand. This industry needs to have its business models analyzed to be in accordance with the needs of its consumers during and after the COVID-19 period.

The essential contribution of this study lies in its empirical research, which makes it possible to improve the identification of the particular impact of preferences for fashion styles on shopping habits during the COVID-19 pandemic, as well as to improve the identification of the concrete impact of fashion style preference on product, promotion and WOM. Besides the conceptual model proposal, another contribution of the study is the model verification based on the answers of 642 respondents, who presented as typical fashion lovers. What also makes this study stand out is the fact that studies on in-depth consumer analysis and fashion style personalities are not publicly available [69].

The presence of empathy and understanding the behavior of target groups of consumers during the COVID-19 pandemic, based on their preferences for specific fashion styles, can lead to improved business in the fashion industry in the post-COVID-19 period. Understanding this and adapting marketing activities, and relying on preferences for certain fashion styles can be very important for managers and fashion companies in general looking to improve their image, reputation and finally, sales during and after the COVID-19 era.

Future directions suggested by this study can also be identified. One direction could be towards segmenting fashion lovers according to their fashion style preferences. Namely, it would be of most interest to see whether and how their shopping habits differ. The segmentation could be done using clustering [70] or biclustering algorithms [71]. Another direction of future studies could be the inclusion of other elements of the fashion marketing mix, such as price, and the inclusion of some other sub-elements of product, promotion and placement and distribution. Additionally, it would be useful to analyze the influence of fashion style preferences on those who do not declare themselves fashion lovers.

Furthermore, although the research was conducted during the pandemic, it would be of great interest to redo the study after the pandemic ends to explore how the model and the consumers' behavior have changed.

**Author Contributions:** Conceptualization, D.N. and M.K.-S.; methodology, D.N.; software, D.N.; validation, D.N. and M.K.-S.; formal analysis, D.N.; investigation, D.N.; resources, D.N. and M.K.-S.; data curation, D.N.; writing—original draft preparation, D.N. and M.K.-S.; writing—review and editing, D.N. and M.K.-S.; visualization, D.N.; supervision, M.K.-S.; project administration, D.N. and M.K.-S.; no funding acquisition; All authors have read and agreed to the published version of the manuscript.

**Funding:** This research received no external funding.

**Informed Consent Statement:** Informed consent was obtained from all subjects involved in the study.

**Acknowledgments:** All individuals included in this section have consented to the acknowledgement.

**Conflicts of Interest:** The authors declare no conflict of interest.

### Appendix A. Parts of the Conducted Survey Related to the Importance of Some Marketing Elements

The following questions were measured on a five-point Likert scale:

| **Fashion style** |
| --- |
| Indicate on a five-point Likert scale your preference of each of the following style: Business, Hip hop, Retro, Sports, Casual, Rock and Roll |
| **WOM** |
| My decision on whether to buy the certain clothes is influenced by the WOM on the model itself |
| My decision on whether to buy the certain clothes is influenced by the WOM on the fashion brand itself |
| My decision on whether to buy the certain clothes is influenced by the WOM on the fashion designer |
| My decision on whether to buy the certain clothes is influenced by the WOM on the quality of the material |
| My decision on whether to buy the certain clothes online will influence my decision to buy clothes that way |
| **Fashion show** |
| My decision on whether to buy the certain clothes is influenced by their appearance in relevant domestic fashion shows |
| My decision on whether to buy the certain clothes is influenced by their appearance in relevant foreign fashion shows |
| Running a fashion show on well-known fashion channels will influence the decision to buy some of the clothes |
| **Promotion** |
| The promotion of the certain clothes on television influences my decision to buy it |
| The promotion of the certain clothes through social networks influences my decision to buy it |
| The promotion of the certain clothes on billboards influences my decision to buy it |
| The promotion of the certain clothes through print media influences my decision to buy it |
| The promotion of the certain clothes through event sponsorships influences my decision to buy it |
| The promotion of the certain clothes through email marketing influences my decision to buy it |
| The promotion of the certain clothes through event marketing influences my decision to buy it |
| The promotion of the certain clothes through internet portals influences my decision to buy it |
| **Positive image** |
| I believe that by buying certain well-known fashion brands, with positive image, I am influencing the creation of a positive image of myself |
| **Covid-19 frequency habit change** |
| During the COVID-19 pandemic, I changed the frequency of buying clothes |
| **Covid-19 shopping habit change** |
| During the COVID-19 pandemic, I changed the habit of the places where I buy clothes |

The following questions were measured on the following scales:

| **Place of distribution** |
| --- |
| I usually buy clothes? |
| 1-In shopping stores, 2-Online, 3-Both ways |
| **Foreign/Domestic** |
| I would rather buy clothes if they were: |
| 1-foreign, 2-domestic (Serbian), 3-I prefer clothes from both origins equally |

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
