# Peer review of "Improving the Economic Sustainability of the Fashion Industry: A Conceptual Model Proposal"

_sustainability, doi:10.3390/su14084726_

Round 1

Reviewer 1 Report

Dear Authors,

Thank you for giving me the opportunity to read your manuscript entitled "Improving the economic sustainability in fashion industry: A conceptual model proposal (sustainability-1634301)".

Introduction: Authors have clearly mentioned the problem, but it would be good if industry report statistics is presented. Line 36 you have mention stock price drop, it would be good if you can present the percentage of sales drop.

Good number of research has been conducted on consumer behavior of fashion apparels. It would be good if you could list some of the research findings or mention the research focus of others. Then you can mention how your research is different from others. Fashion style explanation can be improved.

Line 77 – 79 How can fashion style preference impacts elements of marketing mix, WOM? Not clear?

Your objective of the research is not clearly explained.

Literature Review & Framework

Fashion style preference is one of the factors (line 117) looks fine but what are the other factors? Page 203 - certain elements of fashion marketing mix and WOM not clear and how it can be one of the factors. Your construct for this research needs to be well defined.

Hypothesis not correctly written. No support of any previous research studies done on the variables/ constructs mentioned in the hypothesis.

No clearly justification of the proposed conceptual model.

Research Methodology

Fashion style measurement looks fine. Other variables used in the studies are not clearly mentioned and details about the scale is not presented. Line 297 – 300 not complete.

Findings

Constructs mentioned in Table 2 not explained in the previous sections. I find there is serious flaw in the conceptual model.

Author Response

We firstly want to thank the Reviewer for acknowledging that our paper is of interest for the Journal. We thank the Reviewer for giving us the opportunity to modify the paper so that it lives up to the Journal standards. We would also like to thank the Reviewer for his encouragement that the paper could be accepted for publication after careful modifications.

Comment 1:Introduction: Authors have clearly mentioned the problem, but it would be good if industry report statistics is presented. Line 36 you have mention stock price drop, it would be good if you can present the percentage of sales drop.

Good number of research has been conducted on consumer behavior of fashion apparels. It would be good if you could list some of the research findings or mention the research focus of others. Then you can mention how your research is different from others. Fashion style explanation can be improved.

Line 77 – 79 How can fashion style preference impacts elements of marketing mix, WOM? Not clear?

Your objective of the research is not clearly explained.”

We thank the Reviewer for pointing these suggestions. We also agree that the originality of the paper and the study should have been presented in more detail about percentage of sales drop and consumer behavior of fashion apparels, as well as the difference of our work compared to others and fashion style explanation in more details. We accordingly expanded the Introduction section, where we added several references and needed explanation.

Comment 2:Literature Review & Framework - Fashion style preference is one of the factors (line 117) looks fine but what are the other factors? Page 203 - certain elements of fashion marketing mix and WOM not clear and how it can be one of the factors. Your construct for this research needs to be well defined.

Hypothesis not correctly written. No support of any previous research studies done on the variables/ constructs mentioned in the hypothesis.

No clearly justification of the proposed conceptual model.”

We thank the Reviewer for his/her suggestions. After re-reading the paper, we became aware that we should add more explanation on other factors next to fashion style preference. We hope that the sections that we added more clearly and in more details present the constructs and the aims of the construct. Through the paper we added references which additionally justify our proposed model.

When it comes to hypothesis some corrections are made, however so far to our knowledge, no research was done to link fashion style preference and apparel production and fashion lovers’ behaviour on integrated way. Therefore, we did not add the support of any research studies. We hope that the Reviewer agrees with the changes.

Comment 3:  Research Methodology - Fashion style measurement looks fine. Other variables used in the studies are not clearly mentioned and details about the scale is not presented. Line 297 – 300 not complete.”

We agree that more attention should have been placed on the scales and questions used. In order to make the paper more readable, we added an Appendix with the questions used in the survey at the end of the paper.

Comment 4: Findings - Constructs mentioned in Table 2 not explained in the previous sections. I find there is serious flaw in the conceptual model.”

We agree with the Reviewer that the Table 2 constructs are not explained in the previous sections, so we add the paragraph connected to that.

 We gave our best to incorporate all the suggestions made by the Reviewer.

Reviewer 2 Report

This research addresses a topic of interest, the specific relationships between fashion style preference and certain elements of fashion marketing mix, WOM and consumers behavior through structural equation modeling.

The introduction well contextualizes the subject of the research. Likewise, the literature review includes the most relevant aspects to take into account. References are appropriate and up to date.

The statement of the hypotheses is correct, as well as the proposed conceptual model. A very appropriate methodology has been selected for this research, using SEM. Also, there is a good sample size. In fact, one of the most outstanding aspects of this research is its empirical nature.

The results are clearly presented and the conclusions provide specific recommendations for companies in the sector.

Author Response

We firstly want to thank the Reviewer for acknowledging that our paper addresses a topic of interest, the specific relationships between fashion style preference and certain elements of fashion marketing mix, WOM and consumers behavior through structural equation modeling. We were delighted to read that the Reviewer’s opinion is that the results are clearly presented and the conclusions provide specific recommendations for companies in the sector.

We would also like to thank the Reviewer for his encouragement that the paper could be accepted for publication.

Reviewer 3 Report

Dear authors,

Please take into account the following minor corrections:

Line 150, p. 4: erase “percent”

Please rephrase research questions H1 & H4.  What do you mean with the certain elements of the marketing mix? Be more specific. According to your results you do not measure all Ps of the marketing mix.

Line 309: to confirm or reject…

Lines 419-420: correct the word “buy” instead of “by”

Line 464: “is” instead of “it likely to attract”

Line 452. Is this your Discussion Chapter?

Lines 504 etc: different font color

Lines 510-511: Are there any data for this?

Line 514: I insist of you not using the “marketing mix” as you measured only the P of promotion

Author Response

We want to thank the Reviewer for giving us the opportunity to modify the paper so that it lives up to the Journal standards. We would also like to thank the Reviewer for his encouragement that the paper could be accepted for publication after minor modifications.

We thank the Reviewer for all his/her positive comments on our paper.

Comment 1:Line 150, p. 4: erase “percent””

We thank the Reviewer for pointing this suggestion. We did the correction.

Comment 2:Please rephrase research questions H1 & H4.  What do you mean with the certain elements of the marketing mix? Be more specific. According to your results you do not measure all Ps of the marketing mix.”

We also agree which this suggestion. In the modified version of the paper, we were more specific, so we pointed out that the paper focused on product and promotion as the certain elements of fashion marketing mix. We accordingly expanded the Introduction section, where we added several references including the suggested two references.

Comment 3: “Line 309: to confirm or reject….

Lines 419-420: correct the word “buy” instead of “by”

Line 464: “is” instead of “it likely to attract”

Line 452. Is this your Discussion Chapter?

Lines 504 etc: different font color

Lines 510-511: Are there any data for this?

Line 514: I insist of you not using the “marketing mix” as you measured only the P of promotion”

We thank the Reviewer for pointing these suggestions. We did all the letter corrections, we were precise with the certain elements of marketing mix (product and promotion), also we changed the font color.

When it comes to the Line 452 (Is this your Discussion Chapter?) as the other Reviewer suggested we have removed that whole paragraph (the first paragraph of the section Discussion and managerial implications) as it was misplaced.

We gave our best to incorporate all the suggestions made by the Reviewer.

Reviewer 4 Report

The research addresses a current topic, entering the line of studies that analyze the impacts of the crisis triggered by the Covid-19 pandemic. The literature is consistent and much of it is truly up to date.

The setting of the methodology and the analysis of the results are well addressed. The weaknesses are found, more than anything else, in the lack of clarity of some passages of the work. It is therefore considered necessary to deepen / clarify some points. For example, it is necessary to indicate the period of administration of the survey in consideration of the fact that the Covid-19 pandemic, which has lasted for two years, has gone through different stages of development with different relative impacts, also in relation to the geographical area. It is believed that the period of administration of the survey, and the related restrictive measures may have influenced the responses obtained.

Among the variables considered in the conceptual model, it is not clear how the styles were selected. Likewise, it is not very clear how the 4P model of the marketing mix changes in relation to the fashion sector. The sub-elements of the fashion marketing mix included in the conceptual model seem to refer only to the dimension of communication. In addition, it is suggested to replace the expression, used several times (also in the abstract), "certain elements of fashion marketing mix" as it is too generic. It seems to superficially sketch information relevant to the purpose of the research.

The graphics of the conceptual model should be improved, clarifying the difference between the shapes used to insert the elements (ovals and rectangles) and inserting the hypotheses in correspondence with the arrows that represent the relationships of influence to which they refer. This would facilitate the interpretation of the figure and, in general, the reading of the corresponding section of the text.

Personally, I think it would also be useful to include in the appendix a schematization of the survey, with the propositions / questions posed to investigate each element of the conceptual model.
In the analysis of the results, there is no reference to hypotheses. Clarify which hypotheses are confirmed and which are not, perhaps re-proposing the appropriately modified conceptual model.
In the discussion, it is not very clear why the difference between generalized and specialized marketing activities is introduced. In general, the entire first paragraph of this section does not seem useful for the purpose of the discussion and the objective of the research.
In the final remarks, it would be appropriate to dedicate more space to the usefulness of the research results in post-Covid.

Author Response

We firstly want to thank the Reviewer for acknowledging that our paper is of a current topic and that the setting of the methodology and the analysis of the results are well addressed. We thank the Reviewer for giving us the opportunity to modify the paper so that it lives up to the Journal standards. We are delighted to read that the Reviewer had several very constructive comments that we gladly implemented in our paper.

Comment 1: “The weaknesses are found, more than anything else, in the lack of clarity of some passages of the work. It is therefore considered necessary to deepen / clarify some points. For example, it is necessary to indicate the period of administration of the survey in consideration of the fact that the Covid-19 pandemic, which has lasted for two years, has gone through different stages of development with different relative impacts, also in relation to the geographical area. It is believed that the period of administration of the survey, and the related restrictive measures may have influenced the responses obtained.”

 We thank the Reviewer for pointing this suggestion. We also agree that the originality of the paper and the study should be improved with the sentences which will deepen the specific situation during the pandemic when it was conducted. We accordingly expanded the Research methodology section, where we added deeper explanation about the pandemic period.

Comment 2: “Among the variables considered in the conceptual model, it is not clear how the styles were selected. Likewise, it is not very clear how the 4P model of the marketing mix changes in relation to the fashion sector. The sub-elements of the fashion marketing mix included in the conceptual model seem to refer only to the dimension of communication. In addition, it is suggested to replace the expression, used several times (also in the abstract), "certain elements of fashion marketing mix" as it is too generic. It seems to superficially sketch information relevant to the purpose of the research.”

After re-reading the paper, we became aware that with certain elements of marketing mix we were considering Product (Foreign or domestic brand) and Promotion (Promotion and Fashion show as its sub-element) so we replaced the expression with the “product and promotion”. Also, we additionally add the explanation about how styles were selected, as well as deepen the explanation about fashion marketing mix.

Comment 3: “The graphics of the conceptual model should be improved, clarifying the difference between the shapes used to insert the elements (ovals and rectangles) and inserting the hypotheses in correspondence with the arrows that represent the relationships of influence to which they refer. This would facilitate the interpretation of the figure and, in general, the reading of the corresponding section of the text.”

We agree that more attention should have been placed on the graphical presentation of the conceptual model, so we insert the hypothesis numbers in correspondence with the arrows that represent the relationships of influence to which they referand we add the sentence which clarify the difference between the shapes.

Comment 4: “Personally, I think it would also be useful to include in the appendix a schematization of the survey, with the propositions / questions posed to investigate each element of the conceptual model.”

We completely agree with the comment, so we add the appendix with the schematization of the survey, with the propositions / questions posed to investigate each element of the conceptual model.

Comment 5: “In the analysis of the results, there is no reference to hypotheses. Clarify which hypotheses are confirmed and which are not, perhaps re-proposing the appropriately modified conceptual model. In the discussion, it is not very clear why the difference between generalized and specialized marketing activities is introduced. In general, the entire first paragraph of this section does not seem useful for the purpose of the discussion and the objective of the research.

In the final remarks, it would be appropriate to dedicate more space to the usefulness of the research results in post-Covid.”

We want to thank the Reviewer for this comment. We agree that we should address the confirmation of the hypotheses in the discussion, so we add that part. After re-reding the paper we have also realized that difference between generalized and specialized marketing activities didn’t seem useful for the discussion, so we removed it. Also, in the final remarks we dedicated more space to the usefulness of the research in post-Covid era. 

We gave our best to incorporate all the suggestions made by the Reviewer.

Round 2

Reviewer 1 Report

The updated article looks good. Wish you all the best.